# From Mimivirus to Mirusvirus: The Quest for Hidden Giants

**DOI:** 10.3390/v15081758

**Published:** 2023-08-17

**Authors:** Morgan Gaïa, Patrick Forterre

**Affiliations:** 1Génomique Métabolique, Genoscope, Institut François Jacob, CEA, CNRS, Univ. Evry, Université Paris-Saclay, 91000 Evry, France; 2Research Federation for the Study of Global Ocean Systems Ecology and Evolution, FR2022/Tara GOSEE, 75012 Paris, France; 3Institut de Biologie Intégrative de la Cellule (I2BC), CNRS, Université Paris-Saclay, 91190 Gif-sur-Yvette, France; 4Département de Microbiologie, Institut Pasteur, 75015 Paris, France

**Keywords:** giant viruses, *Nucleocytoviricota*, viral eukaryogenesis, evolution

## Abstract

Our perception of viruses has been drastically evolving since the inception of the field of virology over a century ago. In particular, the discovery of giant viruses from the *Nucleocytoviricota* phylum marked a pivotal moment. Their previously concealed diversity and abundance unearthed an unprecedented complexity in the virus world, a complexity that called for new definitions and concepts. These giant viruses underscore the intricate interactions that unfold over time between viruses and their hosts, and are themselves suspected to have played a significant role as a driving force in the evolution of eukaryotes since the dawn of this cellular domain. Whether they possess exceptional relationships with their hosts or whether they unveil the actual depths of evolutionary connections between viruses and cells otherwise hidden in smaller viruses, the attraction giant viruses exert on the scientific community and beyond continues to grow. Yet, they still hold surprises. Indeed, the recent identification of mirusviruses connects giant viruses to herpesviruses, each belonging to distinct viral realms. This discovery substantially broadens the evolutionary landscape of *Nucleocytoviricota*. Undoubtedly, the years to come will reveal their share of surprises.

## 1. The Maturing Field of Virology

The first modern proposal regarding viruses as infectious agents distinct from bacteria, which emerged in the late 19th century [1,2], laid the foundation of a new biological field, virology, but also regrettably established a biased perspective that persisted for almost a century. Indeed, they characterized viruses as extremely small agents (able to pass through ultrafiltration) propagating infections, i.e., diseases in humans, farming stock, and agriculture, notably. As such, the size criterion, often conflated with simplicity, became fundamental to discriminate cells from viruses, and their study predominantly revolved around the diseases they could cause. The first observations of viral particles (virions) [3,4] described extremely small and static geometric structures able to propagate infections, a portrayal that further led biologists to equate the virus with its virion, a passive protein shell protecting a small genome that could only evolve indirectly through the infection of a cell. Consequently, viruses were primarily perceived as simple entities, outside of life. Even the work of Stanley, who successfully crystallized viral proteins for the first time, earned him a Nobel Prize in Chemistry in 1946, rather than in Physiology or Medicine. In addition, the term “virus” was restricted to those infecting eukaryotes, while those infecting bacteria were called “phages”, mirroring the dichotomic classification of life that became prominent in the 1960s between eukaryotes and prokaryotes [5]. Despite the characterization of the actual tripartite division of life in three cellular domains (Eukarya, Archaea, Bacteria) [6], each associated with a specific portion of the virosphere [7], this nomenclature remained unchanged: to this day, bacterial viruses are still commonly referred to as bacteriophages, a semantic specificity inexistant for the two other domains.

Environmental studies in the late 20th century initiated a transformative shift in our perception of viruses, as they revealed the extreme abundance and diversity of these entities across virtually every biome [8,9,10]. Human and animal tissues, but also oceans, lakes, soil, and every place harboring cellular life, are rich in viruses. In fact, viral particles are estimated to often outnumber Bacteria, the most abundant cellular organisms, by varying magnitudes [10,11], although their resemblance to extracellular membrane vesicles under fluorescence microscopy has at times led to overestimation [12]. From the 1970s, viruses were organized within the Baltimore classification in classes defined only from the genome type (double-stranded DNA, or positively charged single-stranded RNA viruses, for instance), an approach that consequently grouped various evolutionarily unrelated viruses. However, over time, viruses have been slowly categorized from their taxonomy, which is based on gene conservation, sequence, and structure similarities, in the ICTV (International Committee for the Taxonomy of Viruses) classification [13]. The unmatched diversity of viruses, though, complicates this approach, resulting in many unclassified viruses: in 1971, only two families were described, the then-highest taxonomic rank (International Committee on Taxonomy of Viruses (ICTV): https://ictv.global/taxonomy/, accessed on 3 July 2023). In 2022, over 50 years later, viruses were organized into 6 realms, 10 kingdoms, 64 orders, and 169 families [14]; yet, many viruses still evade classification. The extreme abundance and diversity of viruses highlight how an exclusive focus on their potential roles in causing diseases was narrowing our understanding of the position of viruses in the biosphere. Freed from this paradigm, some scientists began to explore the diverse roles that viruses might play, from ecology to evolution, notably in the context of cellular evolution. For instance, more than two decades ago, it was proposed that DNA and DNA replication proteins could have originated from DNA viruses [15,16,17], whereas Takemura and Bell independently developed the viral eukaryogenesis hypothesis, which suggests that complex poxvirus-like DNA viruses could have driven the emergence of the eukaryotic nucleus [18,19].

## 2. The Discovery of Giant Viruses: Defying Conventional Wisdom

The gradual evolution in our understanding of viruses reached a turning point in 2003 with the discovery of the first giant virus, *Mimivirus*, a double-stranded DNA virus of eukaryotes isolated in an amoeba [20]. Mimivirus, with particle size of around 700 nm, proved that viruses can be as large as some small bacteria, shattering the century-old notion of extremely small dimensions. The analysis of Mimivirus’s genome also yielded a major revelation. With a DNA genome of ca. 1.2 Mb encoding nearly 1000 putative genes, including many related to functions previously presumed exclusive to cellular organisms, notably the translational machinery, the giant virus further blurred the boundary between viruses and cells [21]. 

This milestone discovery triggered an intense search for related viruses (see Table 1), uncovering an entire new diversity of giant viruses, from Pandoravirus [22] with a genome reaching 2.5 Mb (similar to some parasitic eukaryotes) to head-tail-shaped tupanviruses [23], or Pithovirus, the largest virus identified so far, boasting a particle size of 1.5 µm [24]. Phylogenetic and phylogenomic analyses showed that, despite their diversity, these viruses were related to each other within the *Nucleocytoviricota* phylum [25] (formerly known as the NucleoCytoplasmic Large DNA Virus assemblage [26]), a group of large dsDNA viruses infecting the entire diversity of eukaryotes [27]. With virion sizes and genome lengths largely overlapping those of cellular organisms, and gene content suggesting previously unknown evolutionary relationships with their hosts, the discovery of giant viruses reinvigorated the debates not only about the origin and evolution of viruses, but also about their very definition, including the controversial query of whether they should be considered as life forms [21,28,29,30,31].

This latter question became particularly prominent with the discovery, in 2008, of viruses able to infect mimiviruses: the virophages [32]. These small viruses (50–75 nm-long particles with dsDNA genomes ranging from 17 to 30 kb) are unable to propagate an infection on their own. Instead, they coinfect their amoebal host alongside a mimivirus. They replicate and form their new particles within the virion factory, the cytoplasmic compartment formed consecutively to most giant viruses’ infection, all at the expense of their associated giant virus’ replication. The discovery of virophages shed light on a new layer of complexity in the interplay between cells and their viruses, and revealed that viruses themselves can be targeted by other viruses. This prompted some to posit that if viruses can become sick, they should be deemed alive [33]. This provocative statement underscores how the demarcation between viruses and their cellular hosts had shifted from a clear delineation to a more complex continuum, necessitating novel concepts and definitions.

**Table 1 viruses-15-01758-t001:** Summary of selected discoveries, observations, and proposals that have marked the field of virology in the last 20 years.

Year	Discovery-Proposals	Notes
2001	Viral eukaryogenesis hypothesis	First proposal of the viral eukaryogenesis hypothesis, suggesting that viruses could have significantly contributed to the emergence of modern eukaryotes, particularly through the formation of the nucleus [18,19].
2003	*Mimivirus*	First giant virus characterized, with a ~1.2 Mb dsDNA genome packed into a ~700 nm capsid (including fibrils) [20]. Its discovery sparked an intense and fruitful search for related giant viruses. They all belong to the *Nucleocytoviricota* phylum, which encompasses several families of large and giant dsDNA viruses infecting the entire eukaryotic diversity [25,26].
2008	Virophages	Small viruses that coinfect their host alongside a mimivirus-like virus, often hindering the latter’s replication [32]. Additional virophages were later discovered [34], and some have been found to integrate their host’s genome [35].
2010	Virocell concept	This concept describes a virus as several transient states: an extracellular particle, a genome, and an infected cell, or virocell [30].
2013–2014	*Pandoravirus* *Pithovirus*	*Pandoravirus* and *Pithovirus* possess the largest viral genome (2.5 Mb for nearly 2500 proteins) and the largest viral particle (1.5 µm), respectively [22,24]. These sizes significantly overlap with those of small cellular organisms.
2018	*Tupanvirus*	A giant virus of the *Imitervirales* order that displays an exceptionally large set of translational apparatus proteins [23]. Remarkably, it has also odd particles: they are the only tailed giant viruses, a morphology reminiscent of some bacteriophages.
2020	Metagenomic abundance	Although several *Nucleocytoviricota* genomes had been characterized through metagenomics, two comprehensive surveys of environmental metagenomes produced many environmental genomes of these viruses, substantially increasing their known diversity [36,37]. These studies facilitate extensive genomic and genetic analyses.
2023	Mirusviruses	A search for *Nucleocytoviricota* and related viruses in oceanic metagenomes from the *Tara Oceans* expedition revealed an entire new group of viruses: the mirusviruses [38]. These viruses have evolutionary relationships with both herpesviruses, with which they share the genes related to the capsid formation, and giant viruses.

## 3. Changing Paradigms

Early on, giant viruses were proposed to constitute a distinct fourth domain of life, apart from Eukarya, Bacteria, and Archaea, with a separate origin (possibly a reduction from an extinct cellular domain) based on phylogenetic inferences of core gene markers [21,39]. These phylogenetic analyses were, however, debated over possible technical biases [28,40,41,42]. Globally though, the fourth-domain hypothesis was essentially disregarded due to larger evolutionary studies that demonstrated the inclusion of *Nucleocytoviricota*—the phylum of large and giant dsDNA viruses—within the *Varidnaviria* viral realm [25,43,44,45]. This realm encompasses other related groups of dsDNA viruses infecting eukaryotes (such as *Adenoviridae* and *Lavidaviridae*), as well as viruses infecting archaea or bacteria (within the *Tectiliviricetes* order). This realm is characterized by the globally shared conservation of the virion morphogenesis module, i.e., the genes related to the formation of the viral particle (the Major Capsid Protein and the packaging ATPase). Importantly, all *Varidnaviria* viruses, except *Nucleocytoviricota*, exhibit more conventional particle and genome sizes. The common origin between giant viruses and smaller *Varidnaviria* viruses hence contradicts the notion of a fourth domain restricted solely to giant viruses, instead suggesting a specific evolutionary path of *Nucleocytoviricota* towards gigantism. This scenario is supported by more recent evolutionary analyses that highlighted several emergences of gigantism within this viral phylum [44,46].

In an effort to propose a comprehensive concept of life that would not exclude viruses as mere inert outsiders, a definition based on specific gene content was suggested in 2008, notably by one of us. This definition posited that viruses would be classified as capsid-encoding organisms, as opposed to ribosome-encoding organisms for the cellular domains [47]. This definition subsequently evolved to refer to virion-encoding/-producing organisms to accommodate for all types of proteinic structures that are used to form virions [48,49], including odd ones such as those of pandoraviruses [22,50]. The intention behind this proposal was to elevate viruses to a comparable status as cells, with a living world divided into two major groups differentiated by single specific features. However, the proposal faced an important challenge as it defined viruses as “organisms”, a term that most biologists have historically reserved for cells [51]. Despite the significant gene content of giant viruses and their susceptibility to infections by smaller viruses, the organismal characterization of viruses remained indeed ambiguous due to their frequent assimilation to their particles. The prevalent particle-centric view of viruses was first challenged by Claudiu Bandea, who in 1983 suggested that viruses should be recognized as organisms with a physiologically active intracellular phase [52], though this notion did not catch much attention at the time. Following the discovery of Mimivirus, the virus–virion paradigm was again challenged by Claverie, who proposed that the virus factory (the complex organelle-like structure that forms in the host’s cytoplasm during the infection cycle) should be considered the actual virus organism [53]. Indeed, this structure, absent in noninfected cells, is strictly specific to the infection by the virus, and a significant part of the viral cycle occurs within, from the replication of the viral genome to the virion assembly. Yet, this proposal, by essence, inherently applies only to viruses whose infection leads to the formation of viral factories (a strategy not confined to giant viruses), thereby excluding a substantial portion of the virosphere, particularly most bacterial and archaeal viruses.

Inspired by a statement from Lwoff over 50 years ago during his 1965 Nobel lecture, wherein he noted that a bacterium infected by a bacteriophage was entirely transformed into a virus factory [54], one of us proposed again that the active form of the virus should not solely be the viral factory but, rather, the entire infected cell, acting as a viral cell, or “virocell” [30,49,55]. The viral factory produced during numerous viruses’ infection could then be likened to a form of viral nucleus of the virocell. Under the “virocell concept”, a virus can be described by several transient states: the virion, corresponding to the inactive extracellular form; the viral genome, which can notably potentially integrate into the host genome; and the infected cell or “virocell”. This latter state could be regarded as the organismal stage of viruses, where the expression of viral proteins profoundly alters (sometimes radically) the host cell’s physiology to ensure the replication of the viral genome and maximize the production of new virions. The term “virocell” is now regularly used in publications that delve into the transformations induced by viral infections [37,56,57] and emphasizes the importance to discriminate virocells from noninfected cells, or ribocells [55], notably in environmental research. Interestingly, the absence of metabolism has long been a distinctive trait of viruses as well as one of the bases to their nonliving classification [28]. However, virocellular metabolism represents the new metabolic state that arises following the expression of virus-encoded metabolic enzymes and/or regulators of cellular pathways [57]. As such, the concept of virocell hence incidentally strengthens the notion of viruses being living organisms. Similarly, viruses were often perceived outside the frame of life owing to an evolution perceived as merely the byproduct of their hosts’ evolution [28]. Incorporating the infected cell into the definition of viruses, however, implies that viruses evolve by themselves. Indeed, during this phage, the viral genome replicates, undergoing variations and selection, often driven by errors in the replication process—as observed in ribocells [58,59]—and possibly leading to new genes progressively originating de novo. Consequently, viral genes likely emerge slowly but continuously within viruses, possibly explaining the large fraction of ORFans in viral genomes and constituting a large reservoir of unknown functions.

In this context, the viral genes identified in cellular genomes and often presumed to have been transferred from another initial host could then actually have a strictly viral origin. This possibility elevates the roles that viruses might have played in the evolution of the cellular domains from mere agents of gene transfers to active sources of genetic novelties [30]. Beyond the strong and permanent pressure of selection they exercise on their hosts, viruses thus instigate variations and serve as a major driver of evolution for life [60]. In that regard, it appears that giant viruses have likely played a significant role in the emergence of modern eukaryotes.

## 4. Giant Viruses and the Viral Eukaryogenesis Hypothesis

Takemura and Bell independently hypothesized over 20 years ago that complex DNA viruses related to Poxvirus might have been involved in the emergence of the eukaryotic nucleus [18,19]. This notion, referred to as the “viral eukaryogenesis hypothesis”, was primarily based on mechanistic similarities between the nucleus and the infection cycle of poxviruses. These commonalities include the uncoupling of transcription and translation, i.e., a replication of the genome protected in an organelle and the transfer of the related mRNA to the cytoplasm where it is translated. Coincidently, poxviruses belong to the *Nucleocytoviricota* phylum, the same group that was shown a few years later to also include the giant viruses [25,26]. Despite not providing clear evidence of an evolutionary relationship between the entire nucleus and the virion factory of *Nucleocytoviricota*, these latter accumulate intriguing elements that strongly suggest one (Figure 1). Structurally, the virion factory of giant viruses indeed exhibits notable analogies with the nucleus [61,62,63,64,65,66]: many viruses recruit parts of the endoplasmic reticulum membrane to form their factory, which occasionally assembles close to the microtubule organizing center, akin to nuclear mitosis. In the case of Mimivirus, the virion factory is even formed directly from the fusion of membrane vesicles that likely originate from invaginations of the nuclear membrane [67]. Beyond these parallels, the genomes of giant viruses have many homologs of eukaryotic-specific genes involved in critical informational processes. These include histones, topoisomerases, helicases, DNA-dependent DNA and RNA polymerases, mRNA capping enzymes, and, in some cases, genes related to the translational apparatus, such as aminoacyl-tRNA synthases (Figure 1) [68,69,70,71,72,73]. Intriguingly, the phylogenetic analyses of several of these genes suggest transfers oriented from viruses to cells, in contrast to the common assumption of viruses merely hijacking genes [46,70,74,75,76]. Importantly, these transfers seem to be particularly ancient, likely predating the emergence of modern eukaryotes. This is notably the case for the DNA-dependent RNA polymerase (RNAP) enzymes [46]. Their two largest subunits carry the structural and functional core of the enzyme, and are universally present in Bacteria, Archaea, and Eukaryotes, as well as in most *Nucleocytoviricota*. Eukaryotes, however, exhibit an additional and enigmatic complexity in the form of three (or more) RNAPs, each specific to a type of RNA (roughly, the RNAP-I is related to rRNA, the RNAP-II to mRNA, and the RNAP-III to tRNA). The in-depth phylogenetic analyses we performed suggest that proto-eukaryotes (the ancestral cellular lineage, regardless of its nature, from which emerged modern eukaryotes) initially possessed a single RNAP that eventually evolved into the RNAP-III. This ancestral enzyme was acquired by the ancestor of *Nucleocytoviricota*, and then retransferred to proto-eukaryotes from a specific lineage of viruses where it had acquired a new specificity, giving rise in proto-eukaryotes to the RNAP-II. The RNAP-I would have emerged through the duplication of one subunit of the ancestral proto-eukaryotic RNAP, fused with a subunit acquired from another group of *Nucleocytoviricota*. A similar scenario of ancient transfer from the ancestor of *Nucleocytoviricota* to proto-eukaryotes was proposed for homologs of the actin genes, potentially leading to conventional eukaryotic actin [75], and for the topoisomerase IIA [76]. These findings substantiated the already-hypothesized ancestry of this group of viruses regarding their hosts, and strongly support a major role of viruses in the emergence of modern eukaryotes, extending beyond the nucleus. Indeed, phylogenomic analyses have revealed that the extent of genes transferred from eukaryotes to *Nucleocytoviricota*, but also from the latter to the former, is particularly important and involves many functional groups [77]. Notably, integrated *Nucleocytoviricota* can constitute a significant portion of their recipients’ gene content, reaching up to 10% in certain green algae [78]. This proportion raises legitimate questions about the extent of the consequences of these past interactions on the biology of their hosts.

Beyond giant viruses, the potential connection between viruses and the eukaryotic nucleus gained further support from the discovery of large bacterioviruses (Jumbo phages, with large genomes) capable of inducing a nucleus-like structure in their bacterial hosts, able to protect the viral DNA against the cellular CRISPR antiviral activity [79,80]. These viral pseudo-nuclei are positioned at a central position within the virocell by a virus-encoded eukaryotic-like tubulin. Additionally, their membrane is also formed by a protein encoded by the virus. Even if these viral nuclei are only analogous to their eukaryotic counterparts, they suggest that the ability to form a nucleus-like organelle can arise in viral genomes. Similarly, the observation of molecular pores traversing the double membrane of coronaviruses’ equivalent of the virion factory (the replication organelle) [81], possibly used to transfer viral mRNA to the cytoplasm, resonates with a potential connection between the eukaryotic nucleus and viruses. These viral pores indeed exhibit significant architectural similarities with the eukaryotic nuclear pores but are encoded by the viruses. While these viral features are not evolutionarily related to the eukaryotic nucleus, they highlight the capacities of viruses to evolve complex features typically associated with cells. Consequently, some traits that are now conventionally considered characteristic of a cellular domain could actually have an origin involving viruses to varying degrees. Nevertheless, no virus seems to have as intricate and ancient an evolutionary relationship with eukaryotes as the giant viruses of the *Nucleocytoviricota* phylum. While it is plausible that smaller viruses have had similarly profound coevolutionary histories with their hosts, these might be not traceable due to the average short genome of viruses. Giant viruses represent perhaps one of the best targets to better understand the coevolution of viruses and their hosts over the history of life.

## 5. Giant Viruses Are Everywhere

The ancestry of *Nucleocytoviricota* and their long-standing interactions with their hosts, from proto-eukaryotes to modern ones, certainly explains their host spectrum, as they collectively infect the entire diversity of the eukaryotic domain [27,82]. This notably translates into their relative ubiquity: since around 2010, it has indeed been shown that *Nucleocytoviricota* viruses can be detected across a large variety of environments [23,83,84,85], particularly aquatic, where they are extremely abundant and diverse [86,87,88,89,90,91,92,93]. Within the *Nucleocytoviricota* phylum, the *Algavirales* and *Imitervirales* orders are particularly represented in these aquatic environments, with viruses notably infecting abundant eukaryotic populations within the plankton, such as *Mamiellales*, or the bloom-forming coccolithophore *Emiliania* and alga *Heterosigma* [87,93,94]. With the rise of viral metagenomics in recent years, the known diversity of *Nucleocytoviricota* has expanded substantially [36,37,38]. Recently, during an investigation of *Nucleocytoviricota* in the metagenomic data generated from the *TARA* Oceans expeditions, we could notably identify and characterize a surprising group of new related viruses: the mirusviruses [38]. These viruses have many genes in common with the *Nucleocytoviricota*, including most of the genes of the core informational module. However, they lack the virion morphogenesis module (i.e., the genes related to the formation of the virions), which is distinctive not only to the *Nucleocytoviricota* but to the entire viral realm they belong to, the *Varidnaviria* realm. This realm of double-stranded DNA viruses is one of very few, with the *Duplodnaviria* realm, notably, that includes viruses infecting the three cellular domains and is hypothesized to have an independent ancient origin. Surprisingly, the mirusviruses possess all the genes from the virion morphogenesis module, but nothing more, of the *Duplodnaviria* realm, to which they then belong in terms of taxonomy despite having many more genes related to the *Nucleocytoviricota*. Indeed, the highest taxonomic ranks of the ICTV classification are based on the rare features shared by various and sometimes distant groups of viruses; for instance, the *Varidnaviria*, the realm *Nucleocytoviricota* belong to, only share a distinctive structure of their capsid protein and a related packaging ATPase [14]. While the *Riboviria* realm (grouping RNA viruses) is defined by its replication machinery, the dsDNA viral realms are characterized by genes of their virion morphogenetic module, often placing an emphasis on the virion structure as the defining element of viruses. The discovery of mirusviruses, constituting a high taxonomic rank (phylum) within the *Duplodnaviria* realm despite having a lot more in common with *Nucleocytoviricota* of the *Varidnaviria* realm, questions the robustness of this classification. The origin of mirusviruses indeed remains enigmatic, whether a giant virus that replaced its virion morphogenesis module with that of a *Duplodnaviria* virus (likely an ancestor of the herpesviruses, the only eukaryotic group of viruses in this realm), or an ancestor of herpesviruses that accumulated many genes from giant viruses, possibly due to close interactions within a similar host. Whichever scenario, the mirusviruses prove that viruses can also bridge unsuspected evolutionary gaps in the viral world. Their discovery may therefore provide fresh insights into our understanding of the origin and evolution of giant viruses, as well as the flows of genes between them, mirusviruses, and their hosts.

Future research will undoubtedly reveal many more unknown viruses that will again change our comprehension of viral evolution and their place in the biosphere.

## Figures and Tables

**Figure 1 viruses-15-01758-f001:**
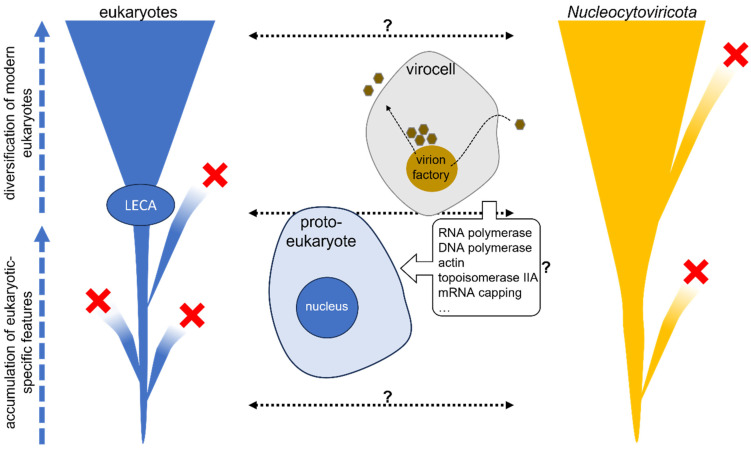
Illustration of the numerous potential gene exchanges between *Nucleocytoviricota* and their hosts, from proto-eukaryotes to modern ones. The two outer tree-like illustrations represent the vertical evolution of eukaryotes and *Nucleocytoviritoca*, while the central part roughly illustrates the parallels between the eukaryotic nucleus and the virion factory of an infected cell (virocell). The viral eukaryogenesis hypothesis posits that the long-lasting interactions between the viruses and their hosts have substantially contributed to the emergence of modern eukaryotes, potentially to their nucleus. Dotted arrows with question marks indicate potential transfers of genes between *Nucleocytoviricota* and their hosts, occurring in both directions and possibly multiple times. These transfers could have included critical core functions that were then acquired by proto-eukaryotes before LECA, the last eukaryotic common ancestor, and been subsequently conserved in all or most modern eukaryotes. The transfers could have involved cellular or viral lineages now extinct (represented by red crosses on the figure).

## Data Availability

No new data were created or analyzed in this study. Data sharing is not applicable to this article.

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
