# Peer review of "From Mimivirus to Mirusvirus: The Quest for Hidden Giants"

_viruses, 2023, doi:10.3390/v15081758_

Round 1

Reviewer 1 Report

The review itself is very interesting and informative. I have no doubt that the authors are experts in their field and that the literature used to write the review is of sufficient quality. However, text is poorly written, which makes it sometimes hard to understand from the first read. But I’m sure that after the language revision this work will be perfectly fit to be published.

The submitted manuscript “From Mimi to Mirus: the quest for hidden giants” requires significant revision in terms of grammatically correct spelling in English. A full revision by a native English speaker is desirable. At the moment, despite the potential usefulness and interest of this scientific review for researchers involved in studies of viruses, it is sometimes difficult to read the material due to the large number of incorrectly composed and poorly written sentences. 

Author Response

We are thankful for the time the referee has invested into reviewing our manuscript. As requested, we have proceeded to the language revision, including with the assistance of a native English speaker, and have accordingly corrected or rephrased several sections. We hope this new version will bring more clarity.

Reviewer 2 Report

The manuscript is a minireview introduced the discovery of giant viruses, diversity, evolution, virus–host and environmental interactions. It is considered worth publishing. However, the manuscript can be further improved by additional revision.

Suggestions for Revision

1. Based on the title, the definition for “Mimi” (mimivirus) and “Mirus” (mirusvirus) should be added. Not so, it may cause puzzlement for readers. In addition, the first giant virus ever discovered, should also give its name (line 65).

2. Figure 1 lacked essential notes. Such as, what does the red “X” and the dotted arrow in the illustration mean?

3. It is not clear that “these viruses” indicated “Giant viruses” or “virus of Nucleocytoviricota” (line 218-220).

4. The earliest and a wide variety of giant viruses all isolated from aquatic organisms. But the related description is too brief, only one sentence “notably aquatic, where they are extremely diverse (line 229)”.

So, it is need to cite relevant references (e.g. Zhang et al, 2022. Recent insights into aquatic viruses: Emerging and reemerging pathogens, molecular features, biological effects, and novel investigative approaches, Water Biology and Security, 2022,1,100062) and supplement corresponding content, to help readers comprehensively understand the background in the evolution and ecology of giant viruses.

5. It is better to provide a table which contains various giant viruses “mimivirus to mirusvirus” and their features, for more prominent and organized.

6. It is better to add a summary section for the review to make the contents and views of the manuscript clearer.

Author Response

We would like to thank the referee for the time invested in the revision of our manuscript and the constructive remarks they made to further improve it.

Please find below the point-by-point response to the specific suggestions, with for each the comment (C) followed by our response (R; in red).

  1. C: Based on the title, the definition for "Mimi' (mimivirus) and "Mirus" (mirusvirus) should be added. Not so, it may cause puzzlement for readers. In addition, the first giant virus ever discovered, should also give its name (line 65).
    R: To avoid confusion for readers, we have opted to change "Mimi" to "Mimivirus" and "Mirus" to "Mirusvirus" in the title directly. This will be more explicit, notably when browsing through a list of titles.
    The name "Mimivirus" has also be included in the section related to its discovery.

  2. C: Figure 1 lacked essential notes. Such as, what does the red "X" and the dotted arrow in the illustration mean?
    R: The referee is correct, and additional notes and explanations have been added to the figure's legend.

  3. C: It is not clear that "these viruses" indicated "Giant viruses" or "virus of Nucleocytoviricota" (line 218-220).
    R: The mention "Nucleocytoviricota viruses" has been included.

  4. C: The earliest and a wide variety of giant viruses all isolated from aquatic organisms. But the related description is too brief, only one sentence "notably aquatic, where they are extremely diverse (line 229)". So, it is need to cite relevant references (e.g. Zhang et al, 2022. [...]) and supplement corresponding content, to help readers comprehensively understand the background in the evolution and ecology of giant viruses.
    R: The referee is right about the importance of giant viruses in aquatic environments, including historically. We have slightly developed this section and included additional references, notably the one suggested by the referee, to better emphasize this aspect.

  5. C: It is better to provide a table which contains various giant viruses "mimivirus to mirusvirus" and their features, for more prominent and organized.
    R: We appreciate this suggestion, and we have now included a new table (Table 1). However, instead of listing their features, we decided to summarize therein the discoveries most relevant to our manuscript.

  6. C: It is better to add a summary section for the review to make the contents and views of the manuscript clearer.
    R: This is a good suggestion that we have carefully considered. We however realized that such a summary section would be a strong repetition of the main text, due to the relative shortness of the manuscript. Instead, the discoveries most relevant to our manuscript are now summarized in the new table.

Reviewer 3 Report

This is a well-written and interesting essay by Gaia and Forterre that will be a nice contribution to the special issue in honor of Dr. Clavierie. The authors discuss some of Claverie's work and it's impact on our understanding of viruses, which makes it a nice fit. I enjoyed the historical elements, such as the earlier work of Claudiu Bandea.

I have no major concerns. It is relatively short and not intended to be comprehensive, and I think that is appropriate given the context and main message. 

Small point - the Gregory et al citation (84) is a bit out-of-place given that they did not discover many giant viruses in this study. I remember looking through the results and seeing maybe 2 contigs that belonged to Phycodnaviridae- so perhaps not a good example to include when discussing the broad distribution of giant viruses. An early paper from Steven Short and Curtis Suttle would perhaps be more appropriate given it was one of the first studies showing these viruses are broadly distributed - https://doi.org/10.1128/AEM.68.3.1290-1296.2002. 

Review written by Frank Aylward

Author Response

We sincerely thank the referee for their time and their comments regarding our manuscript, and we are glad they enjoy reading it.

The referee is correct about the Gregory et al citation. This citation is actually about viruses globally, but surprisingly does not include NCLDVs. We have replaced the reference by the referee's suggestion, and included new ones as well.